# Prevalence and risk factors of hearing loss in the Chinese population aged 45 years and older: Findings from the CHARLS baseline survey

**Xiaoli Xu[1], Gang Sun[2], Deping Sun**ᴵᴰ**[1]\***

**1** Department of Otorhinolaryngology Head and Neck Surgery, The Fourth Clinical College of Chongqing Medical University, Chongqing, People's Republic of China, **2** Department of Otorhinolaryngology Head and Neck Surgery, Shapingba District People's Hospital of Chongqing, Chongqing, People's Republic of China

\* 800306@hospital.cqmu.edu.cn

## Abstract

### Objective

This study aimed to determine the prevalence of hearing loss and identify associated risk factors in a Chinese population aged 45 years and older.

### Study design

This study employed a cross-sectional research design. Data from the 4th wave survey of the China Health and Retirement Longitudinal Study (CHARLS) conducted in 2018 were utilized. Participants were assessed using self-reported questionnaires, and various demographic and comorbidity factors were analyzed to elucidate the risk factors associated with hearing loss.

### Methods

A total of 17,695 individuals from 10,257 households in 450 villages and urban settlements were included in the study. Hearing loss was assessed through self-reported questionnaires. Risk factors, including demographic characteristics and comorbidities, were analyzed to identify associations with hearing loss.

### Results

The study population had a hearing loss prevalence rate of 17.9% (n = 3,179). Regional variations were observed, with highest rates in Chongqing (28.67%), Yunnan (25.12%), and Qinghai (24.36%), and lowest rates in Zhejiang (17.71%), Tianjin (10.56%), and Shanghai (9.26%). Age ≥70 was associated with higher risk (OR = 3, p<0.05), while being female was associated with reduced risk (OR = 0.81, p<0.05). Higher education level showed lower risk (OR<1, p<0.05). Non-agricultural workers had lower risk (OR = 0.67, p<0.05). Fewer social activities were correlated with decreased risk (OR = 0.89, p = 0.024). Ethnic minorities had slightly higher risk (OR = 1.23, p<0.05).

**Data Availability Statement:** We confirm that all data necessary to reproduce our study's results are readily available. The source data is from the China Health and Retirement Longitudinal Study

(CHARLS), which is publicly accessible via their website (http://charls.pku.edu.cn/). Detailed data usage guidelines are provided there. Specific data files and variables used in our analysis are detailed in the Supporting Information section of our manuscript, complete with descriptions and analysis methods. Additionally, the minimal data set for replication is available in Dryad with the DOI: https://doi.org/10.5061/dryad.mpg4f4r85, and is unrestricted for sharing.

**Funding:** The author(s) received no specific funding for this work.

**Competing interests:** The authors have declared that no competing interests exist.

## Conclusion

This study provides valuable insights into the prevalence and risk factors associated with hearing loss in the Chinese population aged 45 years and older. The findings emphasize the importance of early detection and intervention, particularly among older individuals and those residing in specific regions, for effective hearing loss management.

## 1 Introduction

Hearing loss is a significant global health issue, affecting a large number of individuals. In 2019, it was estimated that approximately 1.51 billion to 1.64 billion people worldwide, about one-fifth of the population, experienced hearing loss. These numbers are predicted to rise dramatically by 2050, with an estimated 2.35–2.56 billion individuals suffering from hearing loss. According to the Global Burden of Disease research, hearing loss ranks as the third leading cause of years lived with disability (YLDs) [1].

The impact of hearing loss extends beyond physical implications and affects an individual's quality of life and communication abilities [2, 3]. Studies have found links between hearing loss and cognitive decline [4], increased risk of dementia in older adults, negative emotional experiences [5], and social isolation [6]. Moreover, individuals with hearing loss face higher rates of unemployment compared to those without hearing loss, emphasizing the importance of addressing this issue.

In China, the prevalence of hearing loss is on the rise due to the aging population. Understanding the prevalence and associated risk factors within the Chinese population is crucial for implementing effective prevention and intervention strategies. Previous studies have provided valuable insights into the prevalence of hearing loss among older adults in specific regions of China [7, 8]. However, these studies had limitations in terms of their limited coverage or outdated data.

Self-reported measures of hearing capacity, despite their subjective nature, have proven effective in identifying a wide range of individuals with significant hearing loss. This method offers a more detailed picture of real-world hearing ability, encompassing both the severity of impairment and its perceived effects on daily life. Large-scale epidemiological studies have utilized self-reported hearing loss to provide critical insights into the prevalence and distribution of hearing impairment across populations [9, 10].

To address the limitations of previous research, our study leverages data from the China Health and Retirement Longitudinal Study (CHARLS) to examine the incidence of hearing loss among the middle-aged and elderly. By tapping into the extensive and representative CHARLS dataset, we aim to obtain precise and contemporary data on the prevalence of hearing loss across the entire Chinese population. This research will not only deepen our understanding of the broader implications of hearing loss but also inform the creation of targeted interventions and policies to tackle this critical public health challenge.

## 2 Methods

### 2.1 Ethical approval

This study was conducted in accordance with the Declaration of Helsinki and was approved by the Biomedical Ethics Review Committee of Peking University (IRB00001052-11015). All participants provided informed consent prior to their inclusion in the study. The consent was

obtained in written form, ensuring that participants were fully aware of the study's purpose, procedures, and potential risks. The data were analyzed anonymously to protect the privacy and confidentiality of the participants.

## 2.2 Study population

The study utilized data from the fourth wave of the China Health and Retirement Longitudinal Study (CHARLS), conducted in 2018. The participants included in this wave were individuals aged 45 or above and their spouses, with one participant selected from each household. The CHARLS national baseline survey was first conducted in 2011 and included a total of 17,708 individuals from 10,257 households across 450 villages and urban settlements in 28 provinces [11]. The sampling method used was probability-proportional-to-size (PPS) random sampling, with a stratified multi-stage design based on per capita GDP of urban districts and rural counties [11]. Further details about the CHARLS survey, as well as the data utilized in the present study, can be accessed via the Dryad Digital Repository at the following DOI: https://doi.org/10.5061/dryad.mpg4f4r85.

The study included participants aged 45 years or older. The exclusion criteria consisted of: 1) individuals lacking a hearing questionnaire, 2) Excluding participants with missing survey weight data is crucial to ensure the representativeness of our epidemiological study and maintain the accuracy of our findings, as these weights correct for response bias and ensure our sample accurately reflects the broader population. A total of 17,695 eligible participants' data were included in the statistical analysis. Fig 1 illustrates the flowchart of the data selection process.

## 2.3 Hearing loss

Participants with hearing loss (HL) self-reported their condition in our study. While objective tests such as audiometric testing were not included in the CHARLS survey, previous research has demonstrated the reliability of using questionnaires to identify individuals with hearing problems [12, 13]. Moreover, a validation study conducted in China showed that a single question had a sensitivity of 100% for detecting moderate or higher degrees of hearing impairment in older adults, defined as a pure-tone average at 0.5–4 kHz >40 dB, with a specificity of 84.5% [14]. Additionally, self-reported functional hearing capacity may capture a larger number of people with significant hearing loss and provide a more accurate assessment of real-world hearing ability [15–17].

To address the need for clarity in our assessment of hearing loss, we utilized a self-administered questionnaire that has been previously employed and validated in various research studies. [18–20]. While we acknowledged its use in prior research, we recognize the necessity for a more explicit description of the instrument in our manuscript. The questionnaire, adapted from a validated hearing assessment tool, included the following questions:

1. "Do you ever wear a hearing aid?" with responses categorized as 'Yes' or 'No'.

2. "Would you say your hearing is excellent, very good, good, fair, or poor? Please assess your hearing with a hearing aid if you normally use one, and without if you normally don't." with a Likert scale ranging from 'Excellent' to 'Poor'.

Participants were classified as having hearing impairment based on meeting either criterion: the use of a hearing aid or a rating of 'fair' or 'poor' for their hearing status. This approach ensures a standardized assessment consistent with previously validated methods, allowing for a reliable determination of self-reported hearing loss [12, 13].

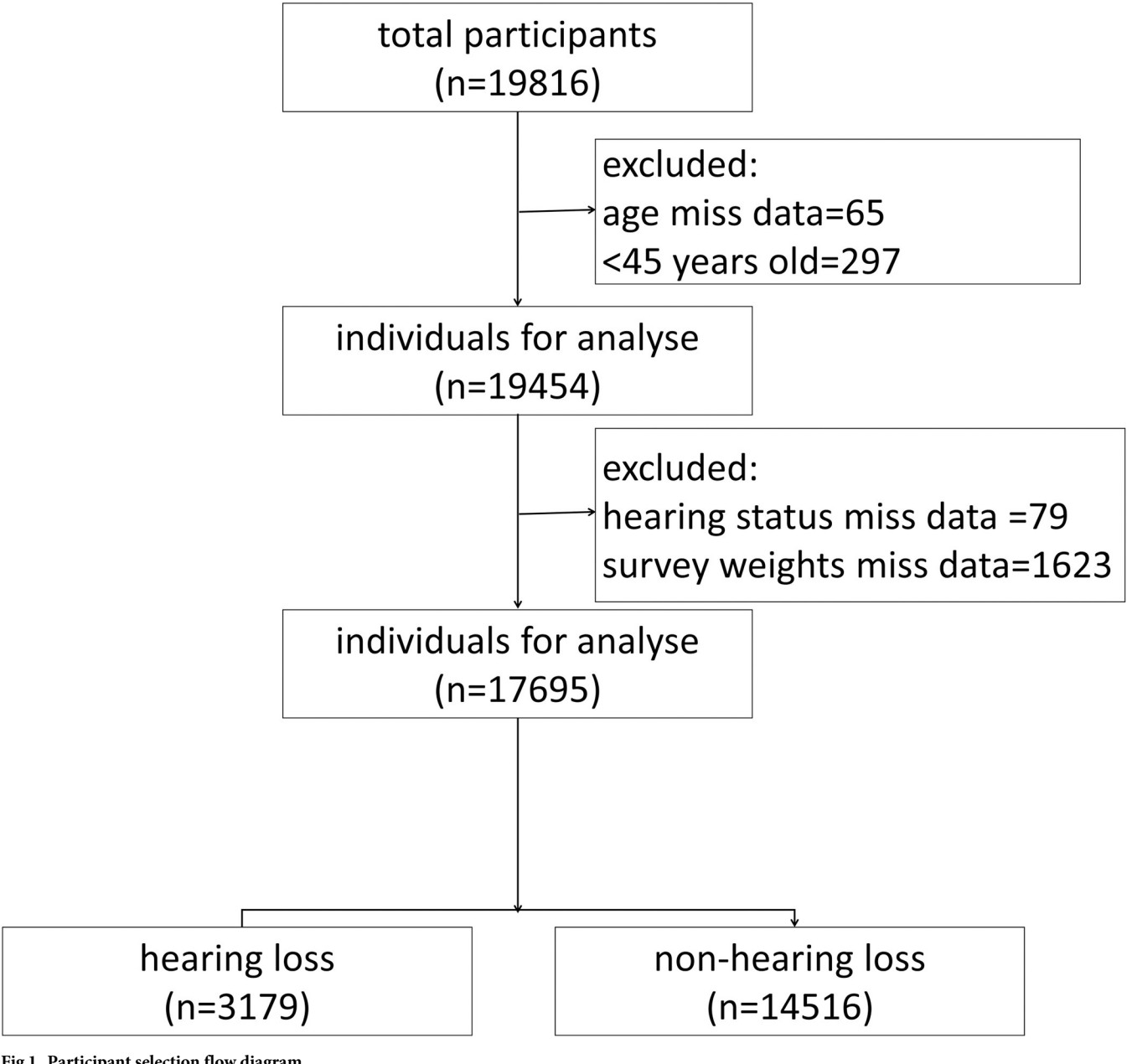

**Fig 1. Participant selection flow diagram.**

This methodology enabled a thorough capture of self-reported hearing capabilities, a significant measure for gauging the impact of hearing loss in everyday scenarios. The questionnaire was meticulously administered via face-to-face interviews by trained investigators, ensuring the precision and uniformity of the collected data.

## 2.4 Covariates

The covariates included in the analysis were: age (45–49, 50–59, 60–69, ≥70), gender (male or female), education level (illiterate, literate, primary, secondary, high school or above), marital status (married or other), residential area (rural or urban), health insurance (covered or not

covered), household size (living alone, two people, three people, ≥four people), occupation type (agricultural, non-agricultural, retired/unemployed), alcohol consumption (drinkers or non-drinkers), current smoking habit (smokers or non-smokers), instrumental activities of daily living (IADLs) difficulty (with or without), any weekly contact with children in person, by phone, mail or E-mail (yes or no), social engagement (Have you done any of these activities in the last month?), nation (Han ethnicity or ethnic minorities).

### 2.5 Handling missing data for categorical variables

To address missing values in categorical variables, we employed the "missing" indicator variable method. This method involves creating a binary variable indicating the presence or absence of missing data. Both the original categorical variable and the "missing" indicator variable were included in the analysis to account for the various categories and the influence of missingness. Treating missing data as an additional category enables examination of its impact and a comprehensive understanding of the data.

### 2.6 Statistical analysis

To address potential bias, the prevalence estimations and risk factors analysis for hearing loss in this study were adjusted by incorporating weights that accounted for the study design, individual weight, and non-response adjustments at both the individual and household levels. All baseline variables in the study were categorical and expressed as absolute counts and percentages. To assess the variations in baseline variables according to the incidence rate of hearing loss, the Kruskal-Wallis test was employed for comparison.

In the multivariate logistic regression analysis, we calculated adjusted odds ratios (ORs) to identify the risk factors associated with hearing loss. The presence of hearing loss was considered the dependent variable in the logistic regression model. The independent variables included various categories such as age, gender, education, living region, health insurance, number of individuals in the household, work status, drinking habits, smoking habits, contact frequency, engagement in social activities, and nationality.

Data cleaning and analysis were conducted using R software (version 4.1.2, R Foundation for Statistical Computing, Vienna, Austria). A significance level of $P < 0.05$ was used to determine statistical significance.

## 3 Results

### 3.1 Demographic characteristics of the participants

Our analysis of 17,695 individuals revealed a 17.9% prevalence of hearing loss. We meticulously examined demographic characteristics, including age, gender, education level, residential area, marital status, and ethnicity. The age distribution across the cohorts was as follows: 9.2% were aged 45–49, 33.5% were 50–59, 34.4% were 60–69, and 22.9% were 70 or older. Regarding education, 24.9% were illiterate, while 11.1% had completed high school or advanced degrees. The majority of participants identified as Han ethnicity, accounting for 92.3% of the study population. These findings are summarized in Table 1.

### 3.2. Prevalence of hearing loss in 2018

The prevalence of hearing loss varied significantly across different demographic factors. Age exhibited a strong association, with higher rates observed as age advanced. Education level also showed a significant correlation. Other factors such as gender, living arrangement, marital status, health insurance coverage, living region, occupation, work status, alcohol consumption,

**Table 1. Summary descriptive table by groups of hearing status.**

| Characteristic | Total | No hearing loss | Hearing loss |
|---|---|---|---|
| Respondents | 17695 | 14516(82.03%) | 3179(17.97%) |
| Age: | | | |
| 45–49 years | 1626(9.2%) | 1471 (10.1%) | 155 (4.88%) |
| 50–59 years | 5929(33.5%) | 5261 (36.2%) | 668 (21.0%) |
| 60–69 years | 6088(34.4%) | 5016 (34.6%) | 1072 (33.7%) |
| ≥70 years | 4052(22.9%) | 2768 (19.1%) | 1284 (40.4%) |
| Gender: | | | |
| Male | 8423(47.6%) | 6961 (48.0%) | 1462 (46.0%) |
| female | 9272(52.4%) | 7555 (52.0%) | 1717 (54.0%) |
| Education: | | | |
| illiterate | 4404(24.9%) | 3272 (22.5%) | 1132 (35.6%) |
| literate | 2854(16.1%) | 2248 (15.5%) | 606 (19.1%) |
| primary education | 5024(28.4%) | 4241 (29.2%) | 783 (24.6%) |
| middle-school | 3441(19.4%) | 2978 (20.5%) | 463 (14.6%) |
| high-school or above | 1972(11.1%) | 1777 (12.2%) | 195 (6.13%) |
| marital status: | | | |
| married | 7258(41.0%) | 5520 (38.0%) | 1738 (54.7%) |
| separated,Divorced or others | 10437(59.0%) | 8996 (62.0%) | 1441 (45.3%) |
| Living region: | | | |
| rural | 10878(61.5%) | 8688 (59.9%) | 2190 (68.9%) |
| urban | 6817(38.5%) | 5828 (40.1%) | 989 (31.1%) |
| Health insurance: | | | |
| no | 658(3.7%) | 516 (3.55%) | 142 (4.47%) |
| yes | 17027(96.2%) | 13991 (96.4%) | 3036 (95.5%) |
| miss | 10(0.1%) | 9 (0.06%) | 1 (0.03%) |
| Household size: | | | |
| living alone | 1543(8.7%) | 1142 (7.87%) | 401 (12.6%) |
| two individuals | 8858(50.1%) | 7231 (49.8%) | 1627 (51.2%) |
| three individuals | 3218(18.2%) | 2718 (18.7%) | 500 (15.7%) |
| ≥four individuals | 4076(23.0%) | 3425 (23.6%) | 651 (20.5%) |
| Work: | | | |
| agricultural | 6344(35.9%) | 5124 (35.3%) | 1220 (38.4%) |
| non-agricultural | 4875(27.6%) | 4400 (30.3%) | 475 (14.9%) |
| retired or no work | 6476(36.6%) | 4992 (34.4%) | 1484 (46.7%) |
| Drinking: | | | |
| no | 11764(66.5%) | 9471 (65.2%) | 2293 (72.1%) |
| yes | 5931(33.5%) | 5045 (34.8%) | 886 (27.9%) |
| Current smoking habit: | | | |
| no | 13041(73.7%) | 10635 (73.3%) | 2406 (75.7%) |
| yes | 3797(21.5%) | 3216 (22.2%) | 581 (18.3%) |
| miss | 857(4.8%) | 665 (4.58%) | 192 (6.04%) |
| IADLs difficulty: | | | |
| no | 11764(66.5%) | 9471 (65.2%) | 2293 (72.1%) |
| yes | 5931(33.5%) | 5045 (34.8%) | 886 (27.9%) |
| FCI: | | | |
| no | 1769(10.0%) | 1396 (9.62%) | 373 (11.7%) |
| yes | 15605(88.2%) | 12869 (88.7%) | 2736 (86.1%) |

*(Continued)*

**Table 1.** (Continued)

| Characteristic | Total | No hearing loss | Hearing loss |
|---|---|---|---|
| miss | 321(1.8%) | 251 (1.73%) | 70 (2.20%) |
| Social engagement: | | | |
| no | 4236(23.9%) | 3672 (25.3%) | 564 (17.7%) |
| yes | 13289(75.1%) | 10728 (73.9%) | 2561 (80.6%) |
| miss | 170(1.0%) | 116 (0.80%) | 54 (1.70%) |
| Nation: | | | |
| Han | 16333(92.3%) | 13437 (92.6%) | 2896 (91.1%) |
| Minorities | 1362(7.7%) | 1079 (7.43%) | 283 (8.90%) |

IADLs: instrumental activities of daily living, FCI: Any weekly contact with children in person, by phone, mail or E-mail/ Frequent Child Interaction

smoking habits, difficulty in instrumental activities of daily living (IADLs), contact with children, engagement in social activities, and ethnicity demonstrated significant variations in hearing loss prevalence,(Table 2). Regional differences were identified, with Chongqing, Yunnan, and Qinghai having the highest rates, and Zhejiang, Tianjin, and Shanghai having the lowest rates (Fig 2).

## 3.3 Sociodemographic, geographic, and lifestyle factors associated with hearing loss

A multivariable logistic regression analysis identified significant risk factors for hearing loss, including age (60–69: OR = 1.57, p<0.05; 70 and above: OR = 3.03, p<0.05), gender (female: OR = 0.88, p<0.05), education level (primary/middle/high school and above: OR<1, p<0.05), work status (non-agricultural: OR = 0.67, p<0.01), without social activity (OR = 0.89, p = 0.024), (OR = 0.89, p = 0.024), and ethnicity (minorities: OR = 1.23, p<0.05) (Fig 3).

## 4 Discussion

According to the study conducted on HCARLS, which involved 17,695 individuals, the overall prevalence of hearing loss was found to be 17.9%. Similarly, the Korea National Health and Nutrition Examination Survey (KNHANES) reported a prevalence of 8% for unilateral hearing loss and 5.9% for bilateral hearing loss among 16,799 aging participants [21]. In the United States, the National Health and Nutrition Examination Survey (NHANES) found that approximately 26% of males and over 20% of females, aged 20 to 80+ years, self-reported trouble hearing [22]. Large-scale population studies such as HCARLS, KNHANES, and NHANES provide comprehensive data on the prevalence of hearing loss across different populations, ensuring reliable and representative results due to their substantial sample sizes. The modest number of participants, a mere 48, reporting hearing aid use in our study indicates a possible underdiagnosis and limited adoption of these devices. This could be due to heightened barriers such as lack of awareness, accessibility issues, or cultural stigma. The small sample underscores the need for further research to uncover the specific challenges faced by hearing aid users and to develop effective intervention strategies.

The study identified certain demographic factors associated with higher rates of hearing loss. These factors include older age groups (particularly 60–69 and ≥70), females, individuals with lower education levels, those who are separated, divorced, or have other marital statuses, and individuals living in rural areas. The highest prevalence rates were observed in the provinces of Chongqing (28.67%), Yunnan (25.12%), and Qinghai (24.36%). Conversely, the lowest

**Table 2. The prevalence of hearing loss by diferent characteristics among people aged 45 and older.**

| Characteristic | Prevalence(%) | 95%CI | P |
|---|---|---|---|
| Respondents | | | |
| Age: | | | <0.001 |
| 45–49 years | 9.533 | 0.08–0.11 | |
| 50–59 years | 11.267 | 0.11–0.12 | |
| 60–69 years | 17.608 | 0.17–0.19 | |
| ≥70 years | 31.688 | 0.30–0.33 | |
| Gender: | | | 0.047 |
| male | 17.357 | 0.17–0.18 | |
| female | 18.518 | 0.18–0.19 | |
| Education: | | | <0.001 |
| illiterate | 25.704 | 0.24–0.27 | |
| literate | 21.233 | 0.20–0.23 | |
| primary education | 15.585 | 0.15–0.17 | |
| middle-school | 13.455 | 0.12–0.15 | |
| high-school or above | 9.8884 | 0.09–0.11 | |
| Marital status: | | | <0.001 |
| married | 23.946 | 0.23–0.249 | |
| separated,Divorced or others | 13.807 | 0.132–0.145 | |
| Living region: | | | <0.001 |
| rural | 20.132 | 0.194–0.209 | |
| urban | 14.508 | 0.137–0.154 | |
| Health insurance: | | | 0.041 |
| no | 21.581 | 0.185–0.249 | |
| yes | 17.831 | 0.173–0.184 | |
| miss | 10 | 0.003–0.445 | |
| Household size: | | | <0.001 |
| living alone | 25.989 | 0.238–0.283 | |
| two individuals | 18.368 | 0.176–0.192 | |
| three individuals | 15.538 | 0.143–0.168 | |
| ≥four individuals | 15.972 | 0.149–0.171 | |
| Work: | | | <0.001 |
| agricultural | 19.231 | 0.1830–202 | |
| non-agricultural | 9.744 | 0.089–0.106 | |
| retired or no work | 22.915 | 0.219–0.24 | |
| Drinking: | | | <0.001 |
| no | 19.492 | 0.188–0.202 | |
| yes | 14.9382 | 0.14–0.159 | |
| Current smoking habit: | | | <0.001 |
| no | 18.45 | 0.178–0.191 | |
| yes | 15.302 | 0.142–0.165 | |
| miss | 22.403 | 0.197–0.253 | |
| IADLs difficulty: | | | <0.001 |
| no | 19.492 | 0.188–0.202 | |
| yes | 14.938 | 0.14–0.159 | |
| FCI: | | | <0.001 |
| no | 21.085 | 0.192–0.231 | |
| yes | 17.533 | 0.169–0.181 | |

*(Continued)*

**Table 2.** (Continued)

| Characteristic | Prevalence(%) | 95%CI | P |
|---|---|---|---|
| miss | 21.807 | 0.174–0.267 | |
| Social engagement: | | | <0.001 |
| no | 13.314 | 0.123–0.144 | |
| yes | 19.272 | 0.186–0.2 | |
| miss | 31.765 | 0.248–0.393 | |
| Nation: | | | 0.005 |
| Han | 17.73 | 0.171–0.183 | |
| Minorities | 20.778 | 0.187–0.23 | |

IADLs: instrumental activities of daily living, FCI: Any weekly contact with children in person, by phone, mail or E-mail.

rates were found in Zhejiang (17.71%), Tianjin (10.56%), and Shanghai (9.26%). These findings highlight the importance of targeted interventions for these high-risk demographic groups.

Multivariate analysis identified several primary risk factors for hearing loss among individuals aged 45 years and above. These factors include age (≥60 years), lower education level (middle-school education and below), being male, and engaging in agricultural work, retirement, or unemployment. The correlation between age and the prevalence of hearing loss aligns with previous studies [23–26], indicating an increased likelihood of hearing loss with

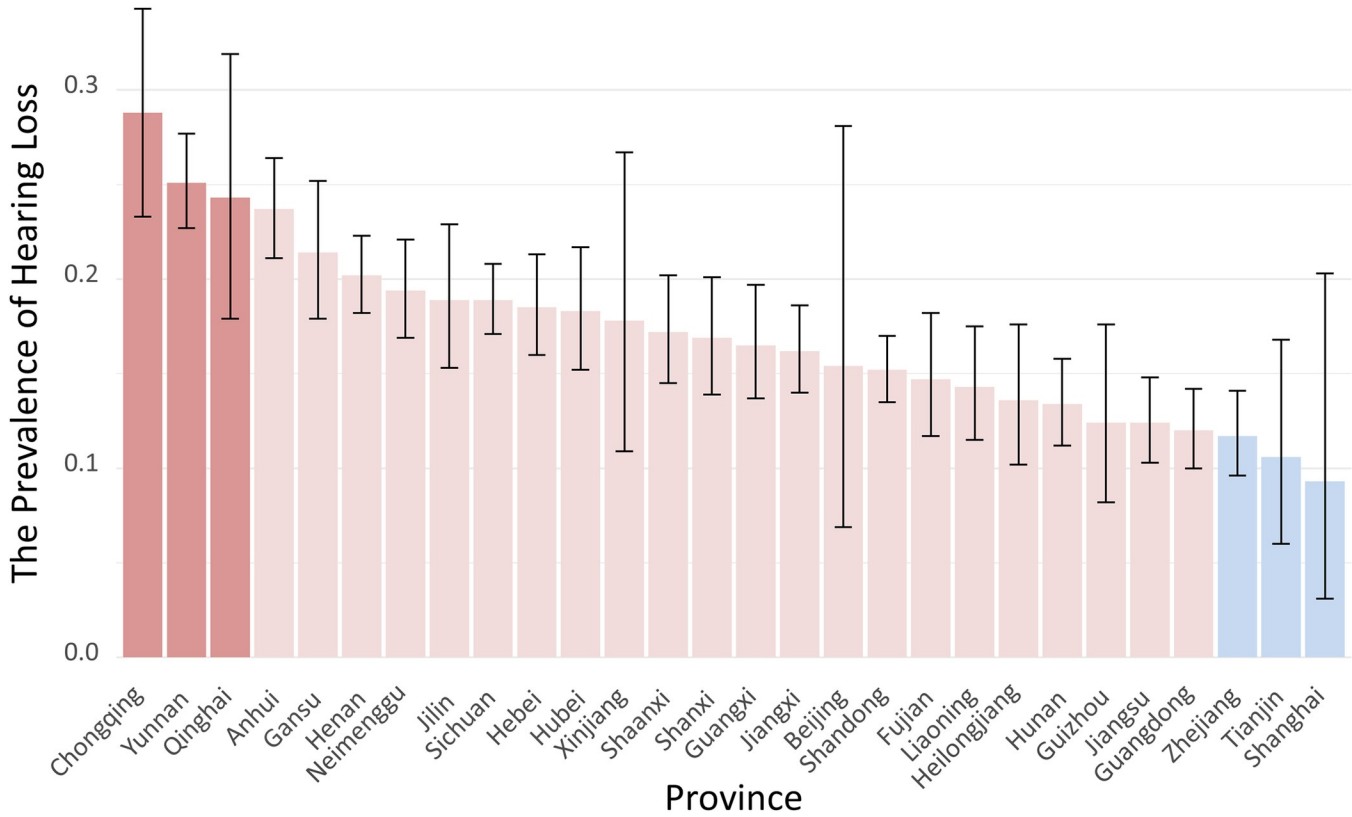

**Fig 2. The prevalence of hearing loss in diffrent province of China.**

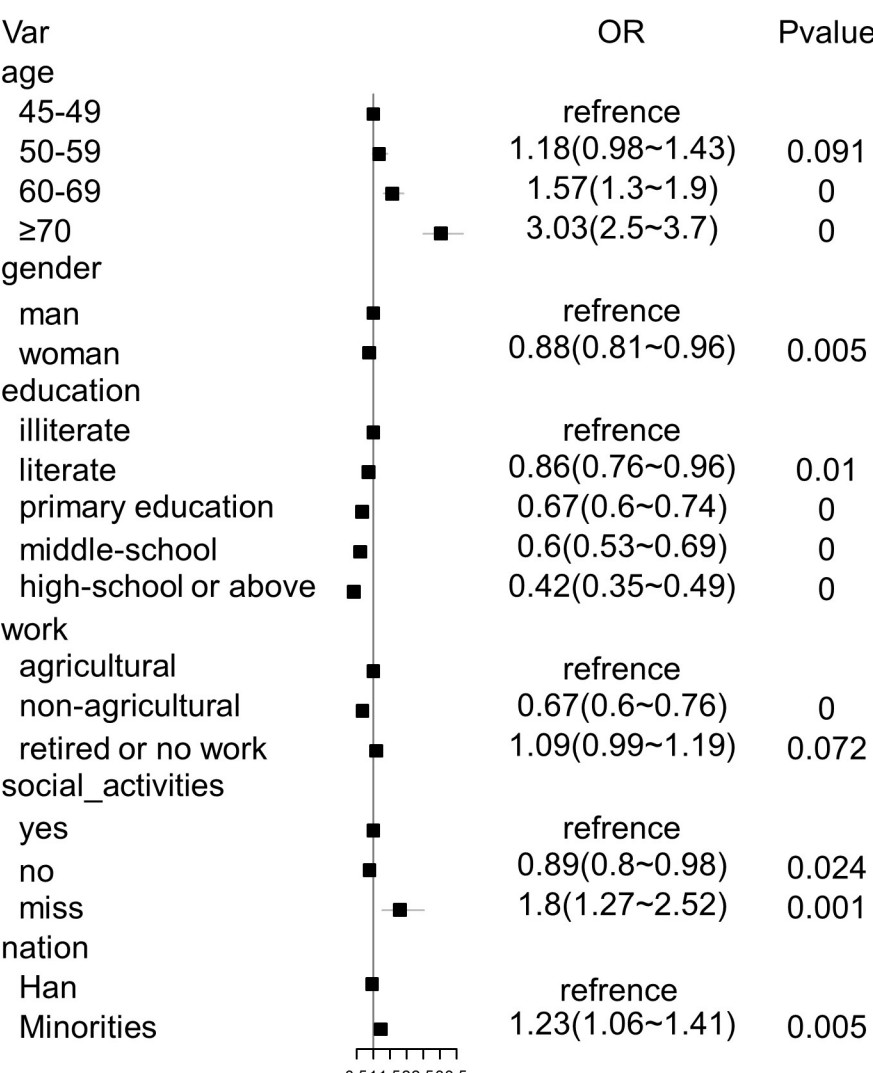

| Var | OR | Pvalue |
|---|---|---|
| age | | |
| 45-49 | refrence | |
| 50-59 | 1.18(0.98~1.43) | 0.091 |
| 60-69 | 1.57(1.3~1.9) | 0 |
| ≥70 | 3.03(2.5~3.7) | 0 |
| gender | | |
| man | refrence | |
| woman | 0.88(0.81~0.96) | 0.005 |
| education | | |
| illiterate | refrence | |
| literate | 0.86(0.76~0.96) | 0.01 |
| primary education | 0.67(0.6~0.74) | 0 |
| middle-school | 0.6(0.53~0.69) | 0 |
| high-school or above | 0.42(0.35~0.49) | 0 |
| work | | |
| agricultural | refrence | |
| non-agricultural | 0.67(0.6~0.76) | 0 |
| retired or no work | 1.09(0.99~1.19) | 0.072 |
| social_activities | | |
| yes | refrence | |
| no | 0.89(0.8~0.98) | 0.024 |
| miss | 1.8(1.27~2.52) | 0.001 |
| nation | | |
| Han | refrence | |
| Minorities | 1.23(1.06~1.41) | 0.005 |

0.5 1 1.5 2 2.5 3 3.5

**Fig 3. Factors associated with hearing loss: Multivariable logistic regression analysis results.**

advancing age. This trend is consistent in other countries [21, 27] as well, emphasizing the importance of understanding the cumulative effects [24, 28] of various factors on hearing loss as individuals age.

Men were found to have a 12% higher risk of hearing loss compared to women, which is consistent with previous research studies [21, 23, 29]. Hormonal differences [30], occupational noise exposure [31–33], and potential genetic factors [34] contribute to this gender disparity.

Low levels of education were identified as a risk factor for hearing loss, with illiterate individuals having a 14% higher risk compared to literate individuals. The risk increases with lower levels of education, indicating the importance of considering socioeconomic disparities [35] and limited access to healthcare [36] in understanding the relationship between education and hearing loss.

While the study suggests that non-agricultural work may be associated with a lower risk of hearing loss, further research is needed to establish causation and explore the complex relationships between occupational factors, lifestyle choices, and hearing outcomes.

We made an intriguing discovery: there was a slight correlation between decreased participation in social activities and a diminished risk of hearing loss (OR = 0.89, p = 0.024). However, additional investigation is required to examine the specific underlying factors.

Additionally, in line with other studies [37], consistent evidence indicates that minorities face a higher risk of hearing loss compared to the Han ethnic group. This increased risk can be attributed to various factors, including genetic variations (such as the GJB2 gene [38]), socioeconomic disparities, limited access to healthcare, cultural practices, environmental factors, and occupational hazards.

Our multivariable analysis points to several key strategies for preventing hearing loss, particularly among those at higher risk. For the elderly, routine hearing screenings coupled with educational programs on hearing preservation are vital. Ensuring gender-neutral access to hearing health resources is essential, as is raising awareness through community programs, especially for individuals with lower levels of education. In the workplace, mandatory hearing protection measures should be emphasized for all, but particularly for non-agricultural workers who may face different noise exposures. Additionally, the study indicates that reduced social activity is associated with a slightly decreased risk of hearing loss, highlighting the need for further research to understand the underlying mechanisms. For ethnic minorities, targeted initiatives that consider cultural nuances are crucial. Implementing these interventions can significantly reduce the prevalence of hearing loss in our targeted populations.

While our study provides valuable insights into the prevalence and risk factors of hearing loss among the Chinese population aged 45 years and older, there are inherent limitations that should be acknowledged. Firstly, the cross-sectional nature of our research design limits our ability to establish causality between the identified risk factors and hearing loss. Secondly, reliance on self-reported data through questionnaires, although validated, may introduce subjective bias, affecting the accuracy of hearing loss prevalence. Thirdly, the generalizability of our findings may be constrained by the regional representation of the CHARLS survey, which may not fully capture the diversity across all areas of China. Additionally, the lack of objective audiometric testing could potentially overlook nuances in the degree and type of hearing loss. Lastly, while we controlled for several variables, there may be residual confounding factors that were not accounted for in our analysis. Despite these limitations, our study offers a robust foundation for further research and underscores the need for early detection and intervention strategies to address hearing loss in the aging population.

The study's robust methodology, characterized by its national scope and diverse participant demographics, significantly enhances the credibility of our findings. Nonetheless, we are mindful of the potential biases arising from the reliance on self-reported data for hearing loss within the CHARLS framework. To mitigate these biases, we advocate for the integration of objective auditory evaluations in subsequent research endeavors, thereby enhancing the precision of prevalence estimates. Our analysis, while revealing several key risk factors, also acknowledges the potential influence of additional variables such as occupational noise exposure, ear diseases, and genetic factors. The inclusion of these elements in future studies is crucial for a more nuanced comprehension of the multifaceted nature of hearing loss and its underlying causes.

## 5 Conclusion

The prevalence of hearing loss is increasing among middle-aged and elderly people in China. Risk factors for hearing loss include age, sex, education level, work, social engagement and ethnicity. These results can help to develop measures to protect hearing and prevent hearing loss.

## Acknowledgments

For the CHARLS data and all the hard work that went into collecting and organizing it, Peking University has our deepest appreciation.

## Author Contributions

**Conceptualization:** Xiaoli Xu, Gang Sun, Deping Sun.

**Data curation:** Xiaoli Xu, Gang Sun, Deping Sun.

**Formal analysis:** Deping Sun.

**Investigation:** Deping Sun.

**Methodology:** Xiaoli Xu, Gang Sun.

**Project administration:** Deping Sun.

**Resources:** Xiaoli Xu, Gang Sun, Deping Sun.

**Supervision:** Deping Sun.

**Validation:** Deping Sun.

**Writing – original draft:** Xiaoli Xu, Gang Sun, Deping Sun.

**Writing – review & editing:** Xiaoli Xu, Gang Sun, Deping Sun.

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
