## [Decision Letter · Decision Letter 0]

29 May 2024

PONE-D-24-06278Prevalence and Risk Factors of Hearing Loss in the Chinese Population Aged 45 Years and Older: Findings from the CHARLS Baseline SurveyPLOS ONE

Dear Dr. sun,

Thank you for submitting your manuscript to PLOS ONE. After careful consideration, we feel that it has merit but does not fully meet PLOS ONE’s publication criteria as it currently stands. Therefore, we invite you to submit a revised version of the manuscript that addresses the points raised during the review process.

**Dear Authors, thank you for submitting your work for consideration. Please, address the comments from the Reviewers. In particular, it is necessary to provide explanations regarding the used questionnaire, I understand that previous work that describes this study's methodology has been cited, but please specify whether the same questionnaire was used and that the mentioned validation studies are for the exact same questionnaire used and presented in your study.**

We look forward to receiving your revised manuscript.

Kind regards,

Irena Ilic, MD, PhD

Academic Editor

PLOS ONE

Journal Requirements:

4. We note that Figure 2 in your submission contain map images which may be copyrighted. All PLOS content is published under the Creative Commons Attribution License (CC BY 4.0), which means that the manuscript, images, and Supporting Information files will be freely available online, and any third party is permitted to access, download, copy, distribute, and use these materials in any way, even commercially, with proper attribution. For these reasons, we cannot publish previously copyrighted maps or satellite images created using proprietary data, such as Google software (Google Maps, Street View, and Earth). For more information, see our copyright guidelines: http://journals.plos.org/plosone/s/licenses-and-copyright.

We require you to either present written permission from the copyright holder to publish these figures specifically under the CC BY 4.0 license, or remove the figures from your submission:

Reviewers' comments:

Reviewer's Responses to Questions

**Comments to the Author**

1. Is the manuscript technically sound, and do the data support the conclusions?

Reviewer #1: Yes

Reviewer #2: Partly

2. Has the statistical analysis been performed appropriately and rigorously? 

Reviewer #1: Yes

Reviewer #2: Yes

3. Have the authors made all data underlying the findings in their manuscript fully available?

Reviewer #1: Yes

Reviewer #2: Yes

4. Is the manuscript presented in an intelligible fashion and written in standard English?

Reviewer #1: Yes

Reviewer #2: Yes

5. Review Comments to the Author

**Reviewer #1:** Dear Author,

Thank you for submitting this paper to the PLOSONE journal.

-I have a question about the criteria to classify hearing impairment: reporting hearing problem, using a HA and having a poor hearing status (how this was evaluated?)

- What some of the suggestion to help those who are at higher risk to develop HL?

-what is your suggestions to reduce the prevalence of HL among the targeted population?

- What are the limitations to the study?

**Reviewer #2: **Title: Prevalence and risk factors of hearing loss in Chinese population aged 45 year and older: Findings from the CHARLS baseline survey

The manuscript describes the findings of a large scale survey of the Chinese population that was conducted on several waves and the results described is from the 4th .

The introduction section requires strengthening and adding more sections about self-reported hearing loss surveys and its use in large population studies. Many statements in the introduction lack referencing. The aims and objectives of study needs to be clearly outlined at the end of the introduction section.

In the methods section the exclusion criteria includes missing information on weights, this needs to be explained to show the significance of this.

The authors described the hearing loss based on self-reported questionnaire, the questionnaire itself was not fully described and was not included in the manuscript, the authors did mention that the questionnaire was previously used and validated in previous studies, but it was not clear if they used the same questionnaires or selected questions of these questionnaires.

The classification of hearing impairment was based on three categories, these categories (needs to be explained to the reader, for example, reporting a hearing problem how does this distinguish normal hearing individuals from individuals with hearing loss? What is meant by having poor hearing status? What criteria did the authors use to classify the patients into this category?

In the results section, Table 1 the terminology needs to be enhanced (for example, man replace with male, woman replace with female), Marry (replace with marital status). Also pay attention to the alignment of the subsections in the table. Similarly, IADL and HwKCNTPMA needs to be replaced with more relevant short terms). Furthermore, in text explanation of the terms is required (for example social engagement).

The results section does not show the results based on the classification of hearing impairment, for instance the participants using hearing aids are the ones with confirmed hearing loss as they would have gone through audiological evaluation prior hearing aid fitting, it would be interesting to see if the findings would be the same if only this group was evaluated?

All of the above suggestions needs to be reflected in the discussion section.

6. PLOS authors have the option to publish the peer review history of their article (what does this mean?). If published, this will include your full peer review and any attached files.

Reviewer #1: No

Reviewer #2: No

---

## [Author Response · Author response to Decision Letter 0]

14 Jul 2024

Thank you for your thorough review and constructive feedback. Below are the streamlined responses to each of your inquiries:

Style and File Naming Requirements:

We have meticulously reviewed and adjusted our manuscript to comply with PLOS ONE's formatting and file naming guidelines, utilizing the provided LaTeX templates.

Data Availability Statement:

We confirm that the "minimal data set" necessary to replicate our study's findings is available in Dryad with the DOI: https://doi.org/10.5061/dryad.mpg4f4r85, and there are no restrictions on data sharing.

Ethics Statement Placement:

The ethics statement has been relocated to the Methods section of the manuscript, ensuring it is included appropriately for publication.

Copyright Concerns for Map Images:

In response to the copyright issue with Figure 2, we have replaced the copyrighted map images with a bar chart that is free from any copyright restrictions.

We trust that these revisions have adequately addressed your concerns and have further strengthened the quality of our submission. We look forward to your continued guidance and feedback.

Sincerely,

Deping Sun

2024-6-28

Dear Reviewer #1,

Thank you for your insightful comments and questions on our manuscript submitted to PLOS ONE. We have taken your feedback into careful consideration and have made the following revisions and clarifications in our manuscript:

Criteria for Classifying Hearing Impairment:

We have clarified our methodology for classifying hearing impairment using a validated self-administered questionnaire. The two key questions regarding the use of hearing aids and self-assessed hearing quality on a Likert scale have been detailed in the revised manuscript to ensure transparency and replicability.

Suggestions for Those at Higher Risk of Developing Hearing Loss (HL):

We have expanded the discussion section to include several strategies aimed at assisting individuals at higher risk of developing HL. These include routine hearing screenings, educational programs, gender-neutral access to resources, community outreach, workplace protection, social engagement, and targeted initiatives for ethnic minorities.

Strategies to Reduce the Prevalence of HL Among the Targeted Population:

The same recommendations listed above are proposed as strategies to reduce the prevalence of HL. These have been concisely summarized and integrated into the discussion to offer a comprehensive approach to prevention.

Limitations of the Study:

We have acknowledged and concisely summarized the key limitations of our study, including the cross-sectional design, reliance on self-reported data, potential lack of regional representation, absence of audiometric testing, and possible residual confounding factors. These limitations are discussed to provide a balanced view of our research findings and to highlight areas for improvement in future studies.

We trust that these revisions adequately address your questions and concerns, and we are grateful for the opportunity to enhance our manuscript based on your feedback.

Sincerely,

Deping Sun

2024-6-28

Dear Reviewer #2,

Thank you for your detailed review and constructive feedback on our manuscript, “Prevalence and Risk Factors of Hearing Loss in the Chinese Population Aged 45 and Older: Findings from the CHARLS Baseline Survey.” We have taken your suggestions seriously and have made the following revisions:

Introduction Enhancement:

We have expanded the introduction to discuss the significance of self-reported hearing loss surveys in large-scale studies and have clarified our study’s aims and objectives at the end of the section. Additionally, we have improved referencing to support our statements.

Exclusion Criteria Explanation:

We have provided a clear explanation of why participants with missing weight data were excluded, emphasizing its importance for the representativeness and accuracy of our findings.

Questionnaire Description:

We have revised the manuscript to confirm that the full, validated questionnaire was used, maintaining consistency with previous research.

Hearing Impairment Classification Clarification:

We have succinctly clarified the classification criteria for hearing impairment in the manuscript, ensuring that the distinctions between categories are clear to the reader.

Terminology and Table Revisions:

Table 1 and the manuscript text have been revised for enhanced clarity and terminology. Gender terms have been standardized, and abbreviations have been defined in the text.

Results Based on Hearing Impairment Classification:

We have included an analysis of the low prevalence of hearing aid use in our discussion, suggesting potential underdiagnosis and barriers to adoption.

We believe these revisions address your concerns and have significantly improved the quality and clarity of our manuscript. We appreciate the opportunity to refine our work based on your comments.

Sincerely,

Deping Sun

2024-6-28

---

## [Decision Letter · Decision Letter 1]

5 Aug 2024

PONE-D-24-06278R1Prevalence and Risk Factors of Hearing Loss in the Chinese Population Aged 45 Years and Older: Findings from the CHARLS Baseline SurveyPLOS ONE

Dear Dr. sun,

Thank you for submitting your manuscript to PLOS ONE. After careful consideration, we feel that it has merit but does not fully meet PLOS ONE’s publication criteria as it currently stands. Therefore, we invite you to submit a revised version of the manuscript that addresses the points raised during the review process.

We look forward to receiving your revised manuscript.

Kind regards,

Irena Ilic, MD, PhD

Academic Editor

PLOS ONE

Journal Requirements:

Additional Editor Comments:

Dear Authors, thank you for submitting your revised version and for providing answers to comments. However, it is not enough to only state that a validated questionnaire was used, one that was used in previous epidemiological studies, but it is necessary to provide citation for those studies, preferably to the validation study.

Reviewers' comments:

Reviewer's Responses to Questions

**Comments to the Author**

1. If the authors have adequately addressed your comments raised in a previous round of review and you feel that this manuscript is now acceptable for publication, you may indicate that here to bypass the “Comments to the Author” section, enter your conflict of interest statement in the “Confidential to Editor” section, and submit your "Accept" recommendation.

Reviewer #2: All comments have been addressed

2. Is the manuscript technically sound, and do the data support the conclusions?

Reviewer #2: Yes

3. Has the statistical analysis been performed appropriately and rigorously? 

Reviewer #2: Yes

4. Have the authors made all data underlying the findings in their manuscript fully available?

Reviewer #2: Yes

5. Is the manuscript presented in an intelligible fashion and written in standard English?

Reviewer #2: Yes

6. Review Comments to the Author

Reviewer #2: All of the comments were adequately addressed by the authors

Typo of CHARLS in the discussion line 202, and line 209.

7. PLOS authors have the option to publish the peer review history of their article (what does this mean?). If published, this will include your full peer review and any attached files.

Reviewer #2: No

---

## [Author Response · Author response to Decision Letter 1]

17 Aug 2024

Dear Irena Ilic, MD, PhD,

We would like to express our gratitude for the opportunity to revise our manuscript entitled "[Manuscript Title]" and for the constructive comments provided by the reviewers. We have taken the feedback seriously and have made the following revisions to our manuscript:

Financial Disclosure Statement: We have included an updated financial disclosure statement in our cover letter as requested, although no changes were necessary.

Figure File Resubmission: We have uploaded our figure files to the Preflight Analysis and Conversion Engine (PACE) and have incorporated the recommended modifications. Specifically, we have replaced Figure 1 and Figure 2 with the revised versions provided by PACE.

Laboratory Protocols: We have deposited our laboratory protocols on protocols.io to enhance the reproducibility of our results, as suggested. The protocols have been assigned their own identifiers (DOIs) for independent citation in the future. We have included the DOI link (https://doi.org/10.5061/dryad.mpg4f4r85) in our methods section, following the guidelines provided.

Reference List: We have meticulously reviewed and updated our reference list to ensure it is complete and accurate. We have confirmed that there are no citations of retracted articles in our manuscript. We have adhered to the "PloS" style for reference formatting and have made sure that all references are up to date.

Citation of Validation Studies: In response to the additional editor comments, we have now provided citations for the validation studies of the questionnaire used in our research. Three relevant studies have been added to the manuscript, ensuring that our use of the validated questionnaire is appropriately referenced.

Thank you again for your guidance. We look forward to the possibility of publication in PLOS ONE.

Sincerely,

Deping Sun

2024-8-17

---

## [Editor Report · Decision Letter 2]

30 Aug 2024

Prevalence and Risk Factors of Hearing Loss in the Chinese Population Aged 45 Years and Older: Findings from the CHARLS Baseline Survey

PONE-D-24-06278R2

Dear Dr. sun,

We’re pleased to inform you that your manuscript has been judged scientifically suitable for publication and will be formally accepted for publication once it meets all outstanding technical requirements.

Kind regards,

Irena Ilic, MD, PhD

Academic Editor

PLOS ONE
---

## [Editor Report · Acceptance letter]

13 Sep 2024

PONE-D-24-06278R2 

PLOS ONE

Dear Dr. sun, 

I'm pleased to inform you that your manuscript has been deemed suitable for publication in PLOS ONE. Congratulations! Your manuscript is now being handed over to our production team.

Kind regards, 

on behalf of

Dr. Irena Ilic 

Academic Editor

PLOS ONE